# Antenatal physical exercise level and its associated factors among pregnant women in Hawassa city, Sidama Region, Ethiopia

**Dereje Zeleke Belachew**[1]*, **Teshome Melese**[2], **Ketemaw Negese**[1], **Gossa Fetene Abebe**[1], **Zemenu Yohannes Kassa**[2]

**1** Department of Midwifery, College of Medicine and Health Sciences, Mizan Tepi University, Mizan, Ethiopia, **2** Department of Midwifery, College of Medicine and Health Sciences, Hawassa University, Hawassa, Ethiopia

* derejezeleke35@gmail.com

**Data Availability Statement:** All relevant data are within the manuscript and its Supporting information files.

## Abstract

### Background

Antenatal physical exercise has roles in health maintenance, prevention, and treatment of disease for pregnant women and fetuses. Different organizations and medical institutions prescribe regular physical exercise during the antenatal period. Despite this, the pregnant populations are less active and decrease their exercise levels during pregnancy than in their non-pregnant state. Therefore, this study aimed to assess antenatal physical exercise level and its associated factors among pregnant women in Hawassa city, Sidama regional state, Ethiopia.

### Methods

Institutional based cross-sectional study design was employed, and 600 study participants were interviewed using a systematic sampling technique from 25th September/2021 to 25th November/2021. Data entry was made using Epi-Data software version 3.1 and exported to SPSS version 25 for analysis. A bivariate logistic regression assessed the association between each independent variable and the outcome variable. Explanatory variables with a p-value of less than 0.25 were a candidate for the multivariable logistic regression. Finally, variables with a p-value of less than 0.05 were declared as statistically significant and reported with their AOR and 95% CI.

### Result

In this study, 25.5% of pregnant women had an adequate practice of antenatal physical exercise while 43.7% of pregnant women had an adequate level of knowledge on antenatal physical exercise An adequate practice of women's antenatal exercise is more likely to occur in women who are exposed to mass media (AOR: 2.43, 95% CI: 1.57, 3.78), Husband college and above educational level (AOR 1.57, 95% CI: 1.05, 6.12), having an adequate level of knowledge (AOR 2.12, 95% CI: 1.13, 3.37), and have of supporting facility (AOR: 2.29, 95% CI: 1.49, 3.51).

**Funding:** The author(s) received no specific funding for this work.

**Competing interests:** The authors have declared that no competing interests exist.

**Abbreviations:** ACOG, American College of Obstetrics and Gynecology; ANC, Antenatal Care; AOR, Adjusted Odds Ratio; APEXs, Antenatal Physical Exercise; CI, Confidence Interva1; COR, Crud Odds Ratio; GDM, Gestational Diabetes Mellitus; GHTN, Gestational Hypertension; MMR, Maternal Mortality Rate; NCD, Non-Communicable Disease; NGO, Non-Governmental Organizations; SPSS, Statistical Packing for Social Sciences; SSA, Sub-Saharan Africa; USA, United States of America; WHO, World Health Organizations.

## Conclusion

In this study, three fourth of the study participants had an inadequate level of practice in antenatal physical exercise than the global standard. It shall be beneficial if the city health administration works towards improving pregnant women's knowledge and practice level on physical exercise by providing information using different media outlets. Healthcare providers should broadcast antenatal physical exercise prescriptions in integration with health-related programs.

## Introduction

In the general population being physically inactive is the fourth leading risk factor of global mortality [1], accounting for 5.3 million (9%) premature mortality [2]. Considering the escalating rate of disease attributable to physically inactiveness, the WHO also developed a global action plan to reduce physical inactivity by at least 10% by 2025 and 15% by 2030 [3]. Especially, for the pregnant population being physically inactive is found to be an independent risk factor for maternal obesity, gestational diabetes mellitus (GDM), and gestational hypertension (GHTN) [4].

Maternal mortality remains a global health problem [2] and the reduction of maternal mortality and morbidity has been identified as a priority area that needs urgent attention by the health sector. As a result, WHO and partners developed a strategy and adopted Sustainable Development Goal target 3.1: to reduce MMR to less than 70 per 100,000 live births by 2030 [2]. In addition to minimizing the direct causes of maternal morbidity; the focus has shifted to correcting the modifiable health risk factors [3, 5]. Physical inactivity during pregnancy is found to be the leading modifiable health risk factor of pregnancy-induced complications [3, 6–8].

Expanding health promotion & preventative services for pregnant women is the main objective of antenatal clinics [9] One of the components of health promotion, and prevention of pregnancy-related complications is physical exercise during pregnancy [10]. A well-established fact in different literature is an active lifestyle modification has significant health benefits for pregnant women and their offspring [9]. Providing effective counseling to pregnant women to continue or commence physical exercise during pregnancy is necessarily improving women's health and it is found to be the initial activity of comprehensive Antenatal care (ANC) [4, 11]. On the other hand, physically inactive pregnant women are associated with increased pregnancy-related complications and lose their lives worldwide [12].

Currently, the World Health Organization (WHO) and the American College of Obstetrics and Gynecology (ACOG) recommend a woman engage in light to moderate aerobic exercise for at least 150 minutes per week (or 30 minutes per day) during pregnancy and postpartum periods if there are no medical or obstetric complications [4, 13]. However, despite clear guidelines and recommendations for antenatal physical exercise set by various institutions, some studies reported that pregnant women are less active than their non-pregnant counterparts and tend to decrease their exercise levels during pregnancy [14, 15]. Most pregnant women do not meet the ACOG and WHO recommended level of physical exercise [16, 17].

In a different study on pregnant women's assessment of antenatal physical exercise, an adequate level of physical exercise in Iran, was found to be 39% [18], in Norway was 14.6% [19], in India 18% [20], and in Nigeria was 10.2% [21]. In Ethiopia, all previous regional studies are

limited to only governmental health facilities, and their findings show that only 8.4% [22], and 11.7% [23] of pregnant women had met the ACOG recommendations [22–24]. This makes to the question of why, considering their socioeconomic, cultural, and educational backgrounds, pregnant women don't exercise. According to the researcher, there is a scarcity of research on the proportion of pregnant women with adequate knowledge and practice in physical activity [22–24].

In addition, there is limited evidence on what factors will affect the knowledge and practice level of physical exercise during pregnancy. Furthermore, factors like mass media exposure, women's partner socio-demographic characteristics, health care provider recommendation, source of information, and availability of supporting facilities have not thoroughly investigated their influence on antenatal physical exercise [22, 23]. Hence, this study aimed to assess antenatal physical exercise level and its associated factors among pregnant women who had ANC in Hawassa city, Sidama Regional State, Ethiopia.

# Method and materials

## Study design and setting

The facility-based cross-sectional study design was employed from 25[th] September to 25[th] November/2021 among pregnant women in Hawassa city, Southern Ethiopia. Hawassa is a city in Sidama regional state located 275 km south of Addis Ababa, the capital city of Ethiopia. According to the 2021 City Health Department estimated report, 394,057 peoples live in Hawassa city, of those, 190,216 were women of reproductive age, and 13,630 of them were pregnant women. The city has eight Sub-city and 32 kebeles, with 86 public and private health institutions. These are one public comprehensive specialized hospital, one general public hospital, one public primary hospital, four private primary hospitals, 11 public health centers, 17 health posts, and 51 private clinics owned by the government. Out of those 21 health institutions (15 public and six private) were given maternal and child health services [7].

## Source of the population

The source populations were all pregnant women who had ANC follow-up at Hawassa city administration health facilities in the year 2021.

## The study populations

The study population was all pregnant women who had ANC follow-up at a selected health facility in Hawassa city administration during the study period.

## The study unit

The study unit was a single woman who systematically selected pregnant women at the selected health facilities in Hawassa city administration during the study periods and fulfilled inclusion criteria.

## Inclusion criteria

Pregnant women who had ANC at a selected health facility during the data collection period were included.

## Exclusion criteria

Pregnant women who were critically ill and unable to communicate, who came to the ANC clinic repeatedly for further consultation or referred cases during data collection time were excluded.

## Sample size, sampling technique, and procedures

The sample size for each specific objective of the study was determined by EPI-info software and the largest was taken as the final sample size (Table 1). The proportion of pregnant women having good knowledge of physical exercise during pregnancy [23] was given the largest sample size (367). Then, using 1.5 design effects and considering a 10% (55) non-response rate, the final sample size was 606 study participants.

A multistage stratified sampling technique was used to select the study participant. First, a simple random method was used to determine five out of fifteen public health facilities and two out of six private health institutions. The sample size for each selected health institution was allocated proportionally based on the previous two-month records of ANC from the ANC logbook in each health facility. Secondly:—eligible pregnant women were recruited at every $k^{th}$ interval using a systematic sampling method in the selected health facility. The $k^{th}$ interval of systematic sampling was determined by dividing the previous two-month client flow by the required sample size from each health institution (K = 3054/606 = 5.04 = 5) (S1 Fig). The lottery method was used to select between 1 and 5. The study unit was selected every five intervals until the desired sample size was attained. In case of unwillingness to participate, the immediate next participant was taken.

## Operational definitions

**Physical exercise.** Is a physical activity consisting of planned, structured, purposive, and repetitive bodily skeletal movements that are done to improve one or more components of physical fitness [4].

**Table 1. Sample size calculation of antenatal physical exercise level and its associated factors among pregnant women in Hawassa city, Sidama Regional State, Ethiopia, 2021.**

| No | Objective | Reference | | | P (%) | d (%) | CI | N |
|---|---|---|---|---|---|---|---|---|
| Women's knowledge on physical exercise | | [23] | | | 39.5 | 5 | 95 | 367 |
| Women's practice on physical Exercise | | [22] | | | 8.4 | 5 | 95 | 177 |
| | | [24] | | | 79.5 | 5 | 95 | 250 |
| | | [23] | | | 11.7 | 5 | 95 | 239 |
| Factors associated with women's knowledge on physical exercise during pregnancy | | PO | P1 | P2 | COR | CI | N | |
| Level of education | | [23] | 80 | 19.6 | 55.38 | 5.1 | 95 | 66 |
| Employment status | | | 80 | 31.52 | 55 | 2.66 | 95 | 154 |
| Parity | | | 80 | 47.87 | 32.37 | 0.52 | 95 | 338 |
| Factors associated with women's practice on physical exercise during pregnancy | | | | | | | | |
| Level of education | | [23] | 80 | 3.92 | 13.84 | 3.94 | 95 | 296 |
| Number of children | | | 80 | 18.78 | 7.19 | 0.34 | 95 | 296 |
| Occupational status | | [22] | 80 | 9.756 | 26.19 | 3.30 | 95 | 194 |

Key:- P- the proportion of pregnant women's knowledge and practice of PExs, d- margin of error, Po- power, P1- percent of the control exposed, P2- percent of the case exposed, COR- Crud odds ratio, CI- confidence interval, N- calculated sample size.

**Antenatal physical exercise.**   In this study antenatal physical exercise means a physical exercise performed during the pregnancy period to improve the physical and psychological well-being of women for labor and prevent pregnancy-induced pathologies.

**Antenatal physical exercise knowledge.**   In the knowledge part of the physical exercise questionnaire, one (1) was awarded for each correct response of the respondent, while each wrong response was assigned zero (0). A summary score (ranging from 0–34) was computed for each participant by summing up the number of items correctly answered. Finally, those participants who answered correctly to 34 knowledge questions scored greater than and equal to the mean value were considered as having adequate antenatal physical exercise knowledge, while those respondents who scored less than the mean value were considered to have inadequate antenatal physical exercise knowledge.

**Antenatal physical exercise practice.**   In the practice part of the antenatal physical exercise questionnaire respondents' response like "not at all" and "yes daily for <30 minutes or weekly < three days" was assigned zero (0) and one (1) was given to those whose response was "yes daily for ≥30 minutes or weekly ≥ three days". Those who exercised at least walking exercise, dancing exercise, cycle driving exercise, breathing exercise, pelvic floor exercise, and ankle and toe exercise antenatal exercise at least three times/150 minutes a week or a minimum of 30 minutes per day was considered as having an adequate antenatal physical exercise practice [4, 13].

**Supportive facility.**   In this study supporting facilities are individuals, teams, or service partners responsible for delivering physical exercise service aids directly or indirectly for the community or individual level.

**Supporting environment.**   In this study, the supporting environment is the nature of the environment founds near working place and leaving the area that contributes to doing physical exercise.

**Mass media exposure.**   Accesses to one of the three media at least once a week (Reading newspaper, watching television, and listening to the radio) [25].

## Data collection tool and methods

The data was collected through an interviewer-administered structure questionnaire, adapted from a previous study conducted by Janakiraman et al. 2021 [23]. Additional variables were added by reviewing relevant literature [4, 21–23, 26–29]. Seven BSc midwives and two health officers who fluently speak and understand both Amharic and the local language (Sidamu Afoo) were assigned as data collectors and supervisors, respectively. The questionnaire consisted of socio-demographic, Obstetrical characteristics, knowledge concerning APEXs, Practice of APEXs during pregnancy, and health-service-related factors.

## Data quality assurance

Initially, the English version (S1 Appendix) of the data collection tool was translated into Amharic language (S2 Appendix) and again back-translated to English to maintain its consistency. Then after, the questionnaire was pretested on 5% (30) of the sample at Adaria general hospital. The reliability of the questions was ascertained using Cronbach's alpha value, and it was 0.906. Before the actual data collection, a conceptual modification was made to the questionnaire to avoid professional jargon, long sentences, and words that might be considered offensive by study participants. The study objective was clarified, and principal investigator gave two days of training to data collectors and supervisors to familiarize them with the data collection tool. The training topics were data confidentiality, responders' rights, informed consent, the techniques of the interview, and filling out the questionnaire. Finally, careful

supervision was carried out every day, and the completeness of the questionnaires was checked daily after the data collection.

## Data management and analysis

Data were coded and entered into a computer using Epi-Data version 3.1. After it had been checked for its completeness, it was then transferred to Statistical Package for Social Sciences (SPSS) software version 25 for analysis. Descriptive statistics like frequency, percentage, and mean were computed, and the results were presented in tables and figures. Bivariate analysis was done, and all explanatory variables with a p-value < 0.25 were entered into the final multi-variable logistic regression to adjust the confounder's effects. Correlation between the independent variable was assessed to test multicollinearity with a variance inflation factor (VIF) of less than 10 for all variables. Furthermore, the Hosmer–Lemeshow goodness-of-fit test was used to assess the model fitness with a p-value greater than 0.05. Variables with a p-value of <0.05 in the multivariable logistic regression model were declared as statistically significant. Finally, the results were reported in terms of AOR with a 95% confidence interval (CI).

## Ethical considerations

Ethical clearance was obtained from Hawassa University College of Medicine and Health Sciences Institutional Review Board (IRB) on 02/09/2021 with reference number IRB/263/13 to conduct the study. An official letter of cooperation from Hawassa University College of Medicine and Health Sciences Department of Midwifery was taken, and then further permission letter was obtained from Hawassa city administrative health department office to the selected Health Facility administrative and department head of the obstetrics ward. After the study protocols had been explained, written informed consent was taken from each participant before an interview.

## Results

### Socio-demographic characteristics of respondents

Among the total 606 pregnant women those who were requested, 600 women consented to the interview to participate in the study giving a response rate of 99%. The mean age of the respondents was 27 years old (SD± 5.413), 77.3%) of them were in the age group of 20–34 years and 84.5% of the respondents were married. More than one-third (41.8%) of the study participants and 28.5% were college or university graduates and attended high school, respectively. Concerning occupation, around 36.5% of women were housewives, while 29.3% were government employers (Table 2).

### Obstetrical characteristics of pregnant women attending ANC in Hawassa city, Sidama region, southern Ethiopia, 2021

Of the total respondents, 63.8% were multigravida, and from that, 75.9% of respondents have not had a history of abortion. More than half of the respondents (56.3%) were in the 2nd trimester of gestation, and 11.3% of the respondents' pregnancy status was unplanned (Table 3).

### Women's knowledge of antenatal physical exercise during pregnancy among pregnant women in in Hawassa city, Sidama Regional State, Ethiopia, 2021

The study assessed pregnant women's knowledge of benefits, contraindications, and time to resume antenatal physical exercise. From the type of physical exercise, about 84.7% of the

**Table 2. Socio-demographic characteristics of pregnant women attending ANC in Hawassa city Health facility, Sidama Regional State, Ethiopia, 2021.**

| Variables | Category | Frequency (%) |
|---|---|---|
| Woman's age | 19 Years old | 65 (10.8) |
| | 20–34 Years old | 464 (77.4) |
| | 35–49 Years old | 71 (11.8) |
| Marital status of women | Unmarried* | 93 (15.5) |
| | Married* | 507 (84.5) |
| Women's occupational status | Housewives | 219 (36.5) |
| | Privet employer | 83 (13.8) |
| | Merchant | 90 (15.1) |
| | Governmental worker | 176 (29.3) |
| | Others* | 32 (5.3) |
| Husband occupation | Farmer | 33 (6.2) |
| | privet employer | 74 (13.8) |
| | Merchant | 190 (35.5) |
| | governmental worker | 206 (38.5) |
| | Others** | 32 (6.0) |
| The educational level of women | no formal education | 45(7.5) |
| | Elementary | 133(22.2) |
| | High school | 171(28.5) |
| | College and university | 251(41.8) |
| Educational Level Of Husband | no formal education | 30(5.6) |
| | Elementary | 88(16.4) |
| | High school | 189(35.4) |
| | College and university | 228(42.6) |
| Monthly income | >1500 ETB | 52(8.7) |
| | 1500–2499 ETB | 45(7.5) |
| | 2500–3499 ETB | 37(6.2) |
| | ≥3500 ETB | 466(77.7) |

Married* married cohabitated, Unmarried* single widowed divorced, others* daily laborer, Farmer and other, others** daily laborer, and other

**Table 3. Obstetrical characteristics of pregnant women in Hawassa city, Sidama Regional State, Ethiopia, 2021.**

| Variables | Category | Frequency (%) |
|---|---|---|
| Gestational age | 1st trimester | 41 (6.8) |
| | 2nd trimester | 338 (56.3) |
| | 3rd trimester | 221 (36.8) |
| Pregnancy status of women | Planned | 532 (88.7) |
| | Unplanned | 68 (11.3) |
| Parity | Primigravida | 164 (27.3) |
| | Multigravida | 383 (63.8) |
| | Grand multigravida | 53 (8.8) |
| Number of children at home | One child | 297 (70.0) |
| | 2–4 Children | 109 (25.7) |
| | ≥5 Children | 18 (4.2) |
| has/had abortion history | Yes | 105 (24.1) |
| | No | 331 (75.9) |

**Table 4. Women's knowledge on the benefit of antenatal physical exercise during pregnancy among pregnant women in Hawassa city, Sidama Regional State, Ethiopia, 2021.**

| Question | Answer | Frequency (%) |
|---|---|---|
| Walking exercise is necessary during pregnancy | False | 92 (15.3) |
|  | True | 508 (84.7) |
| Breathing exercise is essential during pregnancy | False | 294 (49) |
|  | True | 306 (51) |
| Dancing exercise is essential during pregnancy | False | 456 (76) |
|  | True | 144 (24) |
| Running exercise is essential during pregnancy | False | 509 (84.8) |
|  | True | 91 (15.2) |
| Cycle driving exercise is essential during pregnancy | False | 495 (82.5) |
|  | True | 105 (17.5) |
| Pelvic floor exercise is essential during pregnancy | False | 457 (76.2) |
|  | True | 143 (23.8) |
| Ankle and toe exercise is necessary during pregnancy | False | 349 (58.2) |
|  | True | 251 (41.8) |

participants correctly identified walking exercise as essential during pregnancy, followed by breathing exercises at 51% (Table 4).

Regarding pregnant women's knowledge of the benefits of antenatal physical exercise for the mothers during the pregnancy period, from the total respondents, 70.7%, 65.3%, 52.3%and 51.8% of women were correctly knowing antenatal physical exercise during pregnancy can decrease the risk of depression, fatigue and stress, DM, and pre-eclampsia respectively. But 77.2% and 60.8%% of the respondents didn't know antenatal physical exercise during pregnancy can decrease the chance of cesarean section and duration of labor, respectively (Table 5).

Even if more than half (57.8%) of respondents answered physical exercise during pregnancy has a benefit for their fetus. Only 30.8%, and 30.5% of women felt physical exercise during a pregnancy period can reduce the risk of fetal macrosomia and the risk of LBW respectively. On the other hand, most of the respondents 85.3%, 84.2%, and 72.2%, didn't know physical exercise can reduce the risk of abortion, preterm birth, and fetal distress, respectively (Table 6).

**Table 5. Women's knowledge of antenatal physical exercise benefits for the mother among pregnant women in Hawassa city, Sidama Regional State, Ethiopia, 2021.**

| Questions | Answer | Frequency (%) |
|---|---|---|
| Physical exercise during pregnancy has important for the mother | False | 127 (21.2) |
|  | True | 473(78.8) |
| Shorten the duration of labor | False | 365 (60.8) |
|  | True | 235 (39.2) |
| Reduce fatigue and stress during pregnancy | False | 208 (34.7) |
|  | True | 392 (65.3) |
| Reduce depression during pregnancy | False | 176 (29.3) |
|  | True | 424 (70.7) |
| Reduce pre-eclampsia risk during pregnancy | False | 289 (48.2) |
|  | True | 311 (51.8) |
| Reduce the risk of diabetes mellitus during pregnancy | False | 286 (47.7) |
|  | True | 314 (52.3) |
| Reduce the chance of cesarean section | False | 463 (77.2) |
|  | True | 137 (22.8) |

**Table 6. Women's knowledge of the benefit of antenatal physical exercise for their fetus during pregnancy among pregnant women in Hawassa city, Sidama Regional State, Ethiopia, 2021.**

| Questions | Answer | Frequency (%) |
|---|---|---|
| Physical exercise during pregnancy has important for the fetus | False | 253(42.2) |
| | True | 347(57.8) |
| Reduce the risk of preterm birth during pregnancy | False | 505(84.2) |
| | True | 95(15.8) |
| Reduce the risk of abortion during pregnancy | False | 512(85.3) |
| | True | 88(14.7) |
| Reduce the risk of fetal macrosomia during pregnancy | False | 415(69.2) |
| | True | 185(30.8) |
| Reduce the risk of low birth weight during pregnancy | False | 417(69.5) |
| | True | 183(30.5) |
| Reduce the risk of fetal distress during pregnancy | False | 436(72.7) |
| | True | 164(27.3) |

Furthermore, 87.2%, 80.7%, and 73.2% of the respondents knew long-distance running, exhausted tennis bole matching, and exercise in a supine position were unsafe during pregnancy respectively. And also, 86%, 85.3%, 85% and 84.8% of the respondents knew resumption of antenatal physical exercise during muscle weakness, abdominal pain/ painful contraction, vaginal bleeding, headache, and leakage of amniotic fluid is recommended. But only 90.5%, and 54%, of respondents, didn't know about the contraindication of continuing exercising if calf Paine and muscle weakness have happened during exercising, and also 41.8% of respondents considered an exercise in dehydration conditions is safe during the pregnancy period (Table 7).

More than half of the women (55.7%) were reported to haven't participated in regular physical exercise before their pregnancy, and the majority (65.8%) of participants were had never advised about antenatal physical exercise. The most commonly reported source of information about physical exercise was from mass media (34.1%). While 86.8% of pregnant women were not informed about physical exercise during pregnancy by their respective health care providers. The total mean knowledge score of the respondents was found to be 16.96 ± 4.906; it ranges from 6 up to 31. More than half, 56.3% (95% CI, 52, and 60.3) of the respondents had inadequate knowledge of antenatal physical exercise (Table 8).

## Women's practice on antenatal physical exercise during pregnancy among pregnant women attending ANC in Hawassa city, Southern Ethiopia

Regarding the practice of physical exercise, 77% of women practiced at least one type of physical exercise daily and or weekly during the current pregnancy. But only 25.5% (95% CI: 21.5, 29.3) of respondents meet the WHO recommendations. Walking exercise is the most frequently practiced type of exercise, 72.5% followed by ankle and toe exercise 26.5%. While cycle driving (6.7%) and pelvic floor exercise (17.3%) is the least practiced type (S2 Fig).

## Factors associated with women's practice of antenatal physical exercise during pregnancy among pregnant women attending ANC in Hawassa city, Southern Ethiopia

The bivariate analysis of the study revealed that seven explanatory variables were eligible for the multivariate analysis with cut-off points (p < 0.25). These variables are; woman's

**Table 7. Women's knowledge of the contraindications of antenatal physical exercise during pregnancy among pregnant women in Hawassa city, Sidama Regional State, Ethiopia, 2021.**

| Questions | Answer | Frequency (%) |
|---|---|---|
| Exercising in the supine position is safe during pregnancy | True | 161(26.8%) |
| | False | 439(73.2%) |
| Heavy weight lifting exercise is safe during pregnancy | True | 224(37.3%) |
| | False | 376(62.7%) |
| Exercise resulting in strain is safe during pregnancy | True | 176(29.3%) |
| | False | 424(70.7%) |
| Exercise in dehydration conditions is safe during pregnancy | True | 251(41.8%) |
| | False | 349(58.2%) |
| Exhausted tennis bole matching is safe during pregnancy | True | 77(12.8%) |
| | False | 523(87.2%) |
| Long-distance running is safe during pregnancy | True | 116(19.3%) |
| | False | 484(80.7%) |
| If vaginal bleeding is occurring, resumption is essential | False | 88(14.7%) |
| | True | 512(85.3%) |
| If abdominal pain/painful contraction has happened resumption is necessary | False | 84(14.0%) |
| | True | 516(86.0%) |
| If the amniotic fluid is leaked, resumption is mandatory | False | 91(15.2%) |
| | True | 509(84.8%) |
| If the head ache is happening, resumption is best | False | 90(15.0%) |
| | True | 510(85.0%) |
| If chest pain is occurring, resumption is the best decision | False | 112(18.7%) |
| | True | 488(81.3%) |
| If muscle weakness that affects body balance is occurring continuing it will have a risk | False | 543(90.5%) |
| | True | 57(9.5%) |
| If calf pain or body swelling is occurring, the continuation of it is not recommended | False | 324(54.0%) |
| | True | 276(46.0%) |
| Woman's knowledge of physical exercise during pregnancy (**a summary index**) | inadequate knowledge | 338(56.3%) |
| | adequate knowledge | 262(43.7%) |

knowledge level, women's educational level, husband's occupation, husband's educational level, mass media exposure, access to the sport field, and availability of supporting facilities were associated with women's practice level on physical exercise during pregnancy. While as in multivariate analysis, a significant association was found between knowledge and practice of physical exercise. Those participants who had adequate knowledge level had 2.1 times (AOR 2.12, 95% CI: 1.13, 3.37) higher odds of having adequate practice on physical exercise.

On the other hand, women having partners who had college and above educational level were 1.57 times (AOR 1.57, 95% CI, 1.05, 6.12) high odds of having an adequate level of antenatal physical exercise than those who have no formal education. While as in those women who had supporting facilities were had 2.3 times (AOR: 2.29, 95% CI: 1.49, 3.51) and those who were exposed to mass media had 2.4 times (AOR: 2.43, 95% CI: 1.57, 3.78) higher odds of having adequate practiced on physical exercise during pregnancy than those who have not supported by some facility and not exposed to mass media respectively (Table 9).

**Table 8. Health service characteristics and source of information about antenatal physical exercise during pregnancy among pregnant women in Hawassa city, Sidama Regional State, Ethiopia, 2021.**

| Questions | Answer | Frequency (%) |
|---|---|---|
| Is there a supporting facility to do physical exercise | Yes | 261 (43.5) |
| | No | 339 (56.5) |
| Is there a sports field to do physical exercise | Yes | 413 (68.8) |
| | No | 187 (31.2) |
| Do health professionals advise you to do physical exercise | Yes | 205 (34.2) |
| | No | 395 (65.8) |
| Have you got information about physical exercise from the mass media | Yes | 356 (59.3) |
| | No | 244 (40.7) |
| Have you got information about physical exercise from family/ friend | Yes | 121 (20.2) |
| | No | 479 (79.8) |
| Have you got information about physical exercise from books | Yes | 67 (11.2) |
| | No | 533 (88.8) |
| Have you got information about physical exercise from the healthcare provider | Yes | 79 (13.2) |
| | No | 521 (86.8) |
| Have you got information about physical exercise from other sources | Yes | 29 (4.8) |
| | No | 571 (95.2) |

## Discussion

The existing evidence shows that antenatal physical exercise has significant health benefits for pregnant women and their fetuses. Different organizations and medical institutions recommend and prescribe practicing physical exercise regularly. Despite this, especially for the pregnant women physical exercises are poorly practiced.

In this study, from 600 mothers who have ANC follow-up at health institutions in Hawassa city, only 43.7% of women had adequate knowledge of benefits, contraindications, and prerequisites/time to resume physical exercise during pregnancy. This finding was similar to a study conducted in Gondar Ethiopia (39.5%) [23]. The similarity of the finding might be because both studies were done in a country that has a similar health system plan, and a comparable socioeconomic status, for instance, the age of the study participants in both studies was in the age group of 20–24 years, and most of them were college and above educational status.

However, the result of this study was higher than a study done in Zambia (19%) [30]. The higher level of knowledge noticed in this study might be since, in Zambia, the majority of their participants' educational level reports were illiterate/non-formal and primary level of education. On the other hand, study's findings were lower than those in Nigeria (52.4%) [31], and Brazil (65.6%) [32]. The low level of knowledge identified in this study might be due to the low socioeconomic conditions, media coverage, paucity of information regarding physical exercise, and the unavailability of antenatal exercise guidelines at the health institution level.

Generally, in this study women's practice of antenatal physical exercise during pregnancy was inadequate. Over one-fourth (25.5%) of women in the overall sample and 34.4% of women among those who practiced exercise during the antenatal period had adequate practice, were meets the WHO and ACOG recommendations. This result was higher than in studies done in Ethiopia, for instance, Gondar (11.7%) [23] and Tigray (8.4%) [22]. It was also higher than studies done in India (18%) [20] and Nigeria (10.2%) [21]. The higher level of practice noticed in this study might be due to the educational level of the respondents most of them were in high school and above level, and a larger sample size of as many as twice larger. But lower than the study done in Iran (39%) [18]. This might be due to the fact

**Table 9. Bivariate and multivariate analysis of factors associated with women's practice of antenatal physical exercise during pregnancy among pregnant women in Hawassa city, Southern Ethiopia, 2021.**

| Variables | IDP | ADP | COR(95% C.I) | AOR(95% C.I) |
|---|---|---|---|---|
| **women knowledge** | | | | |
| Inadequate | 290(85.8%) | 48(14.2%) | 1 | 1 |
| Adequate | 157(59.9%) | 105(40.1%) | 4.04(2.72, 5.98)** | **2.12(1.13, 3.37)*** |
| **Supporting facility available** | | | | |
| Yes | 164(62.8%) | 97(37.2%) | 2.98(2.04, 4.37)** | **2.29(1.49, 3.51)**** |
| No | 283(83.5%) | 56(16.5%) | 1 | 1 |
| **Husband occupation** | | | | |
| Farmer | 30(90.9) | 3(9.1) | 1 | 1 |
| Privet employer | 56(75.7) | 18(24.3) | 3.21(0.87, 11.79) | 0.97 (0.21, 4.40) |
| Merchant | 142(74.7) | 48(25.3) | 3.38(0.98, 11.57) | 1.10(0.27, 4.54) |
| Governmental worker | 141(68.4) | 65(31.6) | 4.61(1.35, 15.65)* | 1.06(0.25, 4.48) |
| Others** | 22(68.8) | 10(31.3) | 4.54(1.11, 18.48)* | 2.80(0.55, 13.66) |
| **Women education level** | | | | |
| No formal education | 41(91.1) | 4(8.9) | 1 | 1 |
| Elementary | 109(82) | 24(18) | 2.25(0.73, 6.90) | 1.51(0.43, 5.23) |
| High school | 123(71.9) | 48(28.1)) | 4.00(1.35,11.77)* | 2.39(0.72, 7.90) |
| College & above | 174(69.3) | 77(30.7) | 4.53(1.57,13.10)* | 2.19(0.67, 7.17) |
| **Husband educational level** | | | | |
| No formal education | 27(90) | 3(10) | 1 | 1 |
| Elementary | 72(81.8) | 16(18.2) | 2.00(0.54, 7.41) | 1.28 (0.30, 5.34) |
| High school | 141(74.6) | 48(25.4) | 3.06(0.88, 10.55) | 1.17(0.28, 4.75) |
| College & above | 151(66.2) | 77(33.8) | 4.58(1.35,15.60)* | **1.57(1.05, 6.12)**** |
| **Accessible to a sports field** | | | | |
| Yes | 283(68.5 | 130(31.5) | 3.27(2.02, 5.31)** | 1.64(0.83, 3.26) |
| No | 164(87.7) | 23(12.3) | 1 | 1 |
| **Mass media exposure** | | | | |
| Exposed | 121(59%) | 84(41) | 3.28(2.24, 4.80) ** | **2.43(1.57, 3.78)**** |
| Un expose | 326(82.5) | 69(17.5) | 1 | 1 |

*Statically significant at p<0.05.

** Statically significant at p<0.001.

Others** daily laborer and others IDP: Inadequate practice ADP: Adequate practice

that midwives in Iran motivate, counsel, and recommend pregnant women to perform physical exercise.

Factors like women's knowledge level, mass media exposure, husbands' educational level, and availability of supporting facilities are associated with practice level on physical exercise. Those women's who had exposure to mass media had 2.4 times higher odds of having an adequate practice level of antenatal physical exercise when compared with those who were not exposed. The possible reasons might be due to the fact that the use of mass media were reinforce health-seeking behavior by allowing national and local health program planners to reach a large proportion of an intended population within a relatively short time.

In addition, women who had an adequate level of knowledge on physical exercise had 2.1 times higher odds of having an adequate practice leve1 of physical exercise during pregnancy, which was similar to the study done in Ethiopia [23], South Africa [33], and Iran [18]. This is might be because the respondents could perform consistently and frequently with better

recognition if they have basic knowledge of benefits, contraindications, and pre-requests about physical exercise. Another possible reason could be they are better informed and could have more opportunities for information sources about the benefits of physical exercise.

On the hand, women having partners who had college and above education level had 1.7 times higher odds of having an adequate level of practice on antenatal physical exercise during pregnancy compared with those who had no formal education. This is argued with the studies done in Gondar Ethiopia [23], and Kenya [34]. This may be due to the fact that as women's educational status advances, their health seeking-behavior also increases. Moreover, those women and their partners who were more educated might have been interested in asking, reading, listening, and watching information sources related to their health and well-being.

Similarly, pregnant women supported by a supporting facility had 2.3 times high odds of having an adequate practice level. In line with the study result done in [33]. This is might be because women practice exercise regularly when if they were guided, motivated, supervised, and sponsored by an aiding facility or supported by trainer personnel to do physical exercise during pregnancy.

## Limitation of the study

A possible limitation of this study is that the data were collected directly from respondents self-reports only. This will lead to under or over-estimations. The examination of a limited set of influencing factors are some limitations.

## Conclusions

In this study, the level of practice of antenatal physical exercise during the pregnancy period was found to be inadequate compared to the global standard. Also, pregnant woman's knowledge level of antenatal physical exercise during pregnancy was inadequate compared to other similar studies. It was revealed that exposure to mass media, having access to a supportive facility, and having college and a higher degree of husband were all significant predictors of practicing enough antenatal physical exercise during pregnancy.

## Recommendations

Hawassa city health department's office better implement strategies specifically focused on improving the knowledge of pregnant women on physical exercise by delivering pertinent information using different media outlets. The health institutions and health care providers better design health education programs to advise pregnant women and their partner's on antenatal physical exercise by broadcasting appropriate information. NGOs and other stakeholders better to work on the study area maximize maternally and child health services by providing social support to foster antenatal physical exercise conditions for women. Finally, other researchers better obtain information from objective measurement (pedometer or accelerometer) of the average energy expenditure in terms of caloric cost and metabolic equivalents (METs) to estimate the exercise intensity to identify possible factors.

## Supporting information

**S1 Appendix. English version questionnaire.**
(DOCX)

**S2 Appendix. Amharic version questionnaire.**
(DOCX)

**S1 Data. SPSS statistics data.**
(SAV)

**S1 Fig. Schematic presentation of sampling procedure of pregnant women in Hawassa city, Sidama Region, Ethiopia, 2021.**
(TIF)

**S2 Fig. Pregnant women's practice level on antenatal physical exercise during pregnancy at Hawassa city, Sidama Region, Ethiopia, 2021.**
(TIF)

## Acknowledgments

We are gratefully thankful to all woman who participated in this study for their commitment to responding to the interviews. We are also thankful to the Hawassa city health office and health institution for their assistance and permission to undertake the research.

## Author Contributions

**Conceptualization:** Dereje Zeleke Belachew, Zemenu Yohannes Kassa.

**Data curation:** Dereje Zeleke Belachew, Teshome Melese, Ketemaw Negese, Gossa Fetene Abebe, Zemenu Yohannes Kassa.

**Formal analysis:** Dereje Zeleke Belachew, Teshome Melese, Gossa Fetene Abebe, Zemenu Yohannes Kassa.

**Methodology:** Dereje Zeleke Belachew, Ketemaw Negese, Gossa Fetene Abebe.

**Supervision:** Gossa Fetene Abebe.

**Validation:** Dereje Zeleke Belachew, Teshome Melese, Ketemaw Negese.

**Writing – original draft:** Dereje Zeleke Belachew.

**Writing – review & editing:** Dereje Zeleke Belachew, Teshome Melese, Ketemaw Negese, Gossa Fetene Abebe, Zemenu Yohannes Kassa.

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
