## [Decision Letter · Decision Letter 0]

24 Oct 2022

PONE-D-22-22411Antenatal physical exercise level and its associated factors among pregnant women in Hawassa city, Sidama Regional State, Ethiopia, 2021PLOS ONE

Dear Dr. Dereje Zeleke Belachew,

Thank you for submitting your manuscript to PLOS ONE. After careful consideration, we feel that it has merit but does not fully meet PLOS ONE’s publication criteria as it currently stands. Therefore, we invite you to submit a revised version of the manuscript that addresses the points raised during the review process.

We look forward to receiving your revised manuscript.

Kind regards,

Sebsibe Tadesse, PhD

Academic Editor

PLOS ONE

5. Please clarify the title of all the tables.

Reviewers' comments:

Reviewer's Responses to Questions

**Comments to the Author**

1. Is the manuscript technically sound, and do the data support the conclusions?

Reviewer #1: Yes

Reviewer #2: Yes

2. Has the statistical analysis been performed appropriately and rigorously? 

Reviewer #1: Yes

Reviewer #2: Yes

3. Have the authors made all data underlying the findings in their manuscript fully available?

Reviewer #1: Yes

Reviewer #2: Yes

4. Is the manuscript presented in an intelligible fashion and written in standard English?

Reviewer #1: No

Reviewer #2: No

5. Review Comments to the Author

Reviewer #1: Dear PLOS, ONE team of editorials, thank you for the chance given to me to review a manuscript titled “Antenatal physical exercise level and its associated factors among pregnant women in Hawassa city, Sidama Regional State, Ethiopia, 2021”. The following are my comments.

1. Is there recommended and context-based exercise in Ethiopia OR are you assessing it the ambulation of the pregnant women simply?

2. Since the study is the first of its kind it needs exploration of exercise and its levels via qualitative methods before conducting quantitative study.

3. If you have used institutional study population, by default we guess that you have sampling frame. Therefore, is that appropriate to use systematic sampling or random sampling and why you have used systematic sampling?

4. The study had failed to strengthen the adverse consequences of not exercising well and the advantage of antenatal exercise.

5. The methods section needs further concise and brief explanation. For instance, the age of the study population and ethics.

6. Data quality assurance was not well described in three phases.

7. All the manuscript sections should entail their basic components.

8. The statistics need revision e.g., P-value=0.05, fail to present the confidence intervals, the binary and multiple logistic regressions.

Regards,

Reviewer #2: Review comments

I found this article very interesting aiming to describe physical exercise during pregnancy. However, the authors should change the title to “Physical exercise and its associated factors among pregnant women in Hawassa city, Southern Ethiopia.”

Abstract

# Background

The background is too long and shortens it. Antenatal physical exercise ….what does it mean, the health care providers can order the women during ANC visits? It needs clarity.

#Result

There is a punctuation error and you should correct it.

Partners college and above educational level (AOR 1.57, 95% CI: 0.40, 6.12) is not significant. The authors should remove it.

#Introduction

The authors should follow the PLoS one citation style.

Maternal Mortality Rate (MMR) remains a global health problem. It can be changed that maternal mortality remains a global health problem.

A study shows that physical exercise during pregnancy in Iran was (39)?

# Methods

Did the authors use any materials during data collection, if not authors should remove the material in the methods part.

Your study units are your study populations. Therefore, the authors should change the population and study unit.

Exclusion

How to identify critically ill. Did the authors have any checklist to identify critically ill women?

The questionnaire was pretested on 5% (30) of the sample at Adaria general hospital. Where is the city/town? How to control information contamination?

The authors should clearly state the data collectors’ work area.

#Results

It lacks clarity……. majorities (77.3%) of them were in the age group of 20-34 years .

privet employer spelling error

The author should change the income to IMF income classification way.

# Discussion

On the hand, women having partners who had college and above education level had 1.7 times higher odds of having an adequate level of practice on antenatal physical exercise during pregnancy compared with those who had no formal education. Check your table this variable is not significant.

Limitations

The authors remove the cross-study as a limitation.

Recommendations

Your recommendation is too long and shortened based on your pertinent findings in your study area.

6. PLOS authors have the option to publish the peer review history of their article (what does this mean?). If published, this will include your full peer review and any attached files.

Reviewer #1: No

Reviewer #2: No

---

## [Author Response · Author response to Decision Letter 0]

18 Dec 2022

Author response to editor and reviewers 

Dear Editor and Reviewers,

Thank you very much for your email incorporating the insight of the editor and reviewer’s comments. With great respect, we might wish to appreciate your critical and scientific constructive comments that have led to the great improvement of our paper entitled "Antenatal physical exercise level and its associated factors among pregnant women in Hawassa city, Sidama Regional State, Ethiopia, 2021." We’ve tried to refine the article by incorporating the comments given by all reviewers and revised the manuscript accordingly. Our responses are given in a point-by-point manner below for respective reviewers' comments. A separate file of the revised manuscript with track changes, revised manuscript, and point-by-point authors’ responses to the editor and reviewers’ comments was uploaded.

Manuscript ID: PONE-D-22-22411

Manuscript Title: Antenatal physical exercise level and its associated factors among pregnant women in Hawassa city, Sidama Regional State, Ethiopia, 2021

Date: 13/11/22

 

Authors’ response to editorial member comments

Editors comment 1. Please ensure that your manuscript meets PLOS ONE's style requirements, including those for file naming.

Author response 1. The authors acknowledge the editorial member’s important direction and we assure you that the manuscript was prepared well according to The PLOS one manuscript formatting guidelines 

Editors comment 2. We suggest you thoroughly copyedit your manuscript for language usage, spelling, and grammar.

Author response 2. The comment has been fully accepted. The manuscript is revised for spelling, grammar, and English editing issues and important correction is made in the whole manuscript.

Editors comment 3. Please include your full ethics statement in the ‘Methods’ section of your manuscript file. In your statement, please include the full name of the IRB or ethics committee who approved or waived your study, as well as whether or not you obtained informed written or verbal consent. If consent was waived for your study, please include this information in your statement as well.

Author response 3. Many thanks for your suggestion. The ethics statement is stated in the ‘Methods’ section of the manuscript file (See page 9, Line 198-204)

Editors comment 4. In your Data Availability statement, you have not specified where the minimal data set underlying the results described in your manuscript can be found. PLOS defines a study's minimal data set as the underlying data used to reach the conclusions drawn in the manuscript and any additional data required to replicate the reported study findings in their entirety. All PLOS journals require that the minimal data set be made fully available

Author response 4. The comment is fully accepted. All minimal data set used in this study is prepared as supplementary file and are fully available in the manuscript.

Editors (1) comment 5. Please clarify the title of all the tables

Author response 5. The comment is fully accepted and the title of each table was revised.

 

Author’s response to comments from reviewer 1 

Thank you very much for your big constructs. Hereunder are the authors' responses to the comments

Introduction section:

Reviewer (1) comment 1. Is there recommended and context-based exercise in Ethiopia OR are you assessing it the ambulation of the pregnant women simply?

Author response 1. Thanks very much, dear reviewer, for your interesting questions. Yes, in Ethiopia, physical exercise during pregnancy is recommended and it is considered one of the main components of comprehensive antenatal care packages. During ANC visits pregnant women are advised to have daily routine physical exercises for half an hour daily (Ethiopia Ministery of Health. National Antenatal Care Guideline. FMOH. 2022;(February). Also, it was indicated in the operational definition of the main body of the manuscript (See page 7, lines 153-156).

Reviewer (1) comment 2. Since the study is the first of its kind it needs exploration of exercise and its levels via qualitative methods before conducting quantitative study.

Author response 2. Dear reviewer, thank you for your valuable and constructive comments. The minimum acceptable level or recommended level of exercise during pregnancy is 30 mints per day or 3 days per week (WHO. T. Global recommendations on physical activity for health. World Heal Organ. 2010;1999(December):1-6). Also, it was indicated in the operational definition of the main body of the manuscript (See page 7, line 153-156).

Reviewer (1) comment 3. If you have used institutional study population, by default we guess that you have sampling frame. Therefore, is that appropriate to use systematic sampling or random sampling and why you have used systematic sampling?

Author response 3. Dear reviewer thank you a lot. There are some reasons why the Authors of this research preferred to use a systematic sampling technique than simple random to select study participants:-

1. In the ANC clinic there is a continuous case flow. This means, there will be newcomers for the first ANC follow-up, in this case, if we used a simple random sampling technique we missed those newcomers 

2. If we used a simple random sampling technique by using ANC follow-up registration bock as a sampling frame there will be women who completed the visit, were referred and displaced out of the study area, aborted and delivered cases. This means, randomly selected women will not available at the time of data collection.

3. Even if we have used an institutional study population it is difficult to have a sampling frame. Since the study has an ongoing case flow and the final respondent were not determined. In conclusion, the systematic sampling technique is suited for this study and we are preferred to use a systematic sampling technique than random sampling. 

Reviewer (1) comment 4. The study had failed to strengthen the adverse consequences of not exercising well and the advantage of antenatal exercise.

Author response 4. Dear reviewer, many thanks for your comment. The Authors of this research would like to remind you that, from the very beginning the objective of the study was to assess the level of antenatal physical exercise and factors associated with antenatal physical exercise level. Therefore, the adverse consequences of not exercising well and the advantage of antenatal exercise needs additional research. We recommend to researchers to investigate the antenatal physical exercise adverse consequences.

Reviewer (1) comment 5. The methods section needs further concise and brief explanation. For instance, the age of the study population and ethics.

Author response 5. Dear reviewer thanks very so much. The comments has been accepted and it was corrected in the revised manuscript.

Reviewer (1) comment 6. Data quality assurance was not well described in three phases.

Author response 6. Dear reviewer, many thanks for your comment. Your comment contributes a major role in the further improvement of the quality of this manuscript. The comment has been fully accepted and the data quality assurance section of the manuscript is rewritten as per your comment (See page 8, lines 174-184)

Reviewer (1) comment 7. All the manuscript sections should entail their basic components.

Author response 7. Thanks very much for your comment. As per your comment and suggestion, the paragraph of the manuscript is revised. The comment has been accepted and the manuscript sections with its basic components were incorporated into the final revised document

Reviewer (1) comment 8. The statistics need revision e.g., P-value=0.05, fail to present the confidence intervals, the binary, and multiple logistic regressions.

Author response 8. Dear reviewer thanks very much for your insightful, and very constructive comments. We remind you that, COR (95% C.I) represents statistical results at Binary, and AOR (95% C.I) represents multiple logistic statistical results respective to its 95% CI. While P-value for variables in the Bivariable and multivariable logistic regressions was stated as Astrix (*)

Dear reviewer, we will provide further clarification, if the reviewer's concerns are not addressed.

 

Author’s response to comments from reviewer 2 

Thank you very much for your big constructs. Hereunder are the authors' responses to the comments

bstract

# Background

Reviewer (2) comment 1. The background is too long and shortens it.

Author response 1. Dear reviewer thank you for your comments. The comments has been fully accepted and addressed in the revised manuscript 

Reviewer (2) comment 2 Antenatal physical exercise ….what does it mean, the health care providers can order the women during ANC visits? It needs clarity.

Author response 2. Antenatal physical exercise means is a physical exercise performed during the pregnancy period to improve the physical and psychological well-being of women for labor and prevent pregnancy-induced pathologies (ACOG. ACOG Committee Opinion No. 804: Physical Activity and Exercise During Pregnancy and the Postpartum Period. Obstet Gynecol. 2020;137(2):376. doi:10.1097/AOG.0000000000004267). It is well defined and operationalized in the manuscript (See page7 Line153-156)

… the health care providers can order the women during ANC visits? It needs clarity.

Yes, currently in Ethiopia physical exercise during pregnancy is one main component of comprehensive ANC, and it is one core package of ANC intervention for Health promotion, prevention, and treatment of disease during pregnancy. Ethiopian National ANC guideline recommended that health care providers counsel/advise pregnant women to have daily physical exercises for a minimum half hour daily (Ethiopia Ministry of Health. National Antenatal Care Guideline. FMOH. 2022;(February).

#Result

Reviewer (2) comment 3There is a punctuation error and you should correct it.

Author response 3. Dear reviewer thanks for your comment given about the issue. The comments are accepted, the manuscript were carefully revised and after necessary correction was incorporated in the revised manuscript

Reviewer (2) comment 4. Partners college and above educational level (AOR 1.57, 95% CI: 0.40, 6.12) is not significant. The authors should remove it.

Author response 4. Dear reviewer, thank you for your insightful and constructive comments Partners College and above college educational level in this study is a significant variable. Sorry for the editing error, we committed an editing error while writing it. A necessary correction was taken into the revised manuscript (See page 19, table 9)

#Introduction 

Reviewer (2) comment 5. The authors should follow the PLoS one citation style.

Maternal Mortality Rate (MMR) remains a global health problem. It can be changed that maternal mortality remains a global health problem.

Author response 5. Thanks very much for your constructive comments. The comments have been accepted and corrected on the final revised document (See page 3, Line 47).

Reviewer (2) comment 6. A study shows that physical exercise during pregnancy in Iran was (39)?

Author response 6 Thanks very much for your constructive comments. The comments have been accepted and typing error is corrected on the final document (See page 4, Line 72).

# Methods

Reviewer (2) comment 7. Did the authors use any materials during data collection, if not authors should remove the material in the methods part.

Author response 7. Thanks very much for your comments and suggestions. The comments has been fully accepted since the authors of this study were used data collection tool (questioners) to collect data. The manuscript is corrected as per the comment (See page 8, Line 165).

 

Reviewer (2) comment 8. Your study units are your study populations. Therefore, the authors should change the population and study unit.

Author response 8. Many thanks for your comment. To clarify more, totally we have 21 health institutions (15 public and six private) health institutions gives maternal and child health service. Those are our source of the population. Initially, five out of 15 public health facilities and two out of six private health institutions were selected by using a simple random sampling method. Those who were in randomly selected health institutions are our study populations. After the sample size was allocated proportionally based on the previous two-month records of ANC from the ANC logbook in each health facility eligible pregnant women were recruited every kth interval by using a systematic sampling method which is our study unit. Since we have a secondary sampling technique the study populations and study units are different (See page 6-7, Line 126-135).

Exclusion

Reviewer (2) comment 9. How to identify critically ill. Did the authors have any checklist to identify critically ill women?

Author response 9. Dear reviewer thanks very much for your insightful, and very constructive comments. The comments has been accepted and necessary correction is made into the revised document, since the Authors have not used any checklist to assess their illness status (See page 6, Line 112).

Reviewer (2) comment 10. The questionnaire was pretested on 5% (30) of the sample at Adaria general hospital. Where is the city/town? How to control information contamination? The authors should clearly state the data collectors’ work area.

Author response 10. Dear reviewer thanks very much for the interesting questions. Adaria general hospital is one of the health facilities found in the city administration and it is one of the sources of the population but not the study population. Concerning information contamination, initially, Adaria general hospital was not selected through simple random sampling, and after that pretest was done 14 days prior to the actual data collection time. Moreover, this study participant who came for referral reasons or further consultation were not interviewed, which is stated under the section of exclusion criteria (See Fig 1)

#Results

Reviewer (2) comment 11. It lacks clarity……. majorities (77.3%) of them were in the age group of 20-34 years privet employer spelling error. The author should change the income to IMF income classification way.

Author response 11. Dear respected reviewer thanks very much for your insightful comments. A correction was made on the revised document (See page 9, Lines 206-213). The Author of this research .preferred to use the Ethiopian taxation and demographic health survey income classifications system in order to compare with the similar regional study finding 

# Discussion

Reviewer (2) comment 12. On the hand, women having partners who had college and above education level had 1.7 times higher odds of having an adequate level of practice on antenatal physical exercise during pregnancy compared with those who had no formal education. Check your table this variable is not significant.

Author response 12. Dear reviewer, thank you for your concerns Partners College and above educational level is a significant variable. Sorry for the editing error, we committed an editing error while writing it. A necessary correction was taken into the revised manuscript (See page 19, table 9)

Limitations

Reviewer (2) comment 13. The authors remove the cross-study as a limitation (See page 22, Lines 355-357)

Author response 13. Dear reviewer thanks for your comment given about the issue. The comments are accepted. The manuscript is corrected based on your recommendation

Recommendations

Reviewer (2) comment 14. Your recommendation is too long and shortened based on your pertinent findings in your study area.

Author response 14. Dear reviewer thanks for your comment given about the issue. The comments are accepted. The manuscript recommendation section is concisely rewritten based on your suggestions (See page22, Line363-374)

---

## [Editor Report · Decision Letter 1]

26 Dec 2022

Antenatal physical exercise level and its associated factors among pregnant women in Hawassa city, Sidama Region, Ethiopia.

PONE-D-22-22411R1

Dear Dr. Dereje Zeleke Belachew,

We’re pleased to inform you that your manuscript has been judged scientifically suitable for publication and will be formally accepted for publication once it meets all outstanding technical requirements.

Kind regards,

Sebsibe Tadesse, PhD

Academic Editor

PLOS ONE

---

## [Editor Report · Acceptance letter]

19 Apr 2023

PONE-D-22-22411R1 

Antenatal physical exercise level and its associated factors among pregnant women in Hawassa city, Sidama Region, Ethiopia. 

Dear Dr. Belachew:

I'm pleased to inform you that your manuscript has been deemed suitable for publication in PLOS ONE. Congratulations! Your manuscript is now with our production department. 

Kind regards, 

on behalf of

Dr. Sebsibe Tadesse 

Academic Editor

PLOS ONE